# Joint Modal Alignment and Feature Enhancement for Visible-Infrared Person Re-Identification

**DOI:** 10.3390/s23114988

**Published:** 2023-05-23

**Authors:** Ronghui Lin, Rong Wang, Wenjing Zhang, Ao Wu, Yihan Bi

**Affiliations:** 1School of Information and Cyber Security, People’s Public Security University of China, Beijing 100038, China; nuaappsuclrh@126.com (R.L.); wenjingppsuc@126.com (W.Z.);; 2Key Laboratory of Security Prevention Technology and Risk Assessment of Ministry of Public Security, Beijing 100038, China

**Keywords:** person re-identification, visible-infrared images, modal alignment, multi-grain, data augmentation

## Abstract

Visible-infrared person re-identification aims to solve the matching problem between cross-camera and cross-modal person images. Existing methods strive to perform better cross-modal alignment, but often neglect the critical importance of feature enhancement for achieving better performance. Therefore, we proposed an effective method that combines both modal alignment and feature enhancement. Specifically, we introduced Visible-Infrared Modal Data Augmentation (VIMDA) for visible images to improve modal alignment. Margin MMD-ID Loss was also used to further enhance modal alignment and optimize model convergence. Then, we proposed Multi-Grain Feature Extraction (MGFE) Structure for feature enhancement to further improve recognition performance. Extensive experiments have been carried out on SYSY-MM01 and RegDB. The result indicates that our method outperforms the current state-of-the-art method for visible-infrared person re-identification. Ablation experiments verified the effectiveness of the proposed method.

## 1. Introduction

Person re-identification (person re-id) is a technique for retrieving and matching specific pedestrian images across cameras [1,2] and is crucial for intelligent surveillance. Visible-infrared person re-identification is a subtask of person re-id that aims to facilitate cross-modal retrieval and matching between visible and infrared person images. This technology has gained significant attention in the field of intelligent analysis for infrared cameras, which are widely deployed in cities to enhance citizen security. Given its capacity to accurately and efficiently recognize individuals in both day and night settings, visible-infrared person re-id has the potential to significantly enhance public safety.

Traditional person re-id has prioritized addressing challenges such as viewpoint changes, pose changes, and occlusions [3,4,5]. However, visible-infrared person re-id presents a distinct challenge that necessitates a shift in focus towards modal alignment. This task is complicated by the significant differences between visible and infrared images, with visible images containing richer color and texture information compared to infrared images. The existing methods for modal alignment, which aim to reduce the modal discrepancy, can be divided into two categories. The first category focuses on the feature extraction process by using carefully designed network architectures and loss functions to extract modal alignment features [6,7,8,9,10,11]. However, these methods often suffer from significant design bias and complex network structures, which may limit their transferability. The second category attempts to reduce the modal discrepancy at the pixel level. For example, GAN-based methods [12,13] transform visible images into an intermediate representation between visible and infrared modals, but at the cost of high computational complexity and additional noise. Alternatively, Ye et al. [14] transformed visible images into a random single-channel RGB image or retained the original image. This method effectively reduces the modal difference, but part of the color information is lost, and more diverse training samples will be introduced, which makes it difficult for the model to converge.

In addition to the modal alignment issue, we observed that visible-infrared person re-id, as an extension of the traditional person re-id, is also a multi-grain instance recognition task, where different granularity features in the person image, such as the person’s appearance profile, sleeve length and buttons on the clothing, can be used as critical information to identify the pedestrian. Therefore, for visible-infrared person re-id, it is insufficient to solely focus on modal alignment; it is also essential to enhance the features of the person image by improving the multi-grain information component. However, there are currently only a limited number of methods available for this purpose [15].

This paper proposes the Joint Modal Alignment and Feature Enhancement (MAFE) network for visible-infrared person re-id. For better modal alignment, the grayscale conversion operation for visible images was introduced together with the data augmentation method in method [14], which is named Visible-Infrared Modal Data Augmentation (VIMDA); at the same time, the probability ratio of image conversion and retention in VIMDA was tested to obtain the best performance. In addition, Margin MMD-ID Loss [16] was also introduced to align the distribution of visible images and infrared images and to improve the convergence of the model. Together, the integration of VIMDA and Margin MMD-ID Loss is collectively referred to as Modal Align Operations (MAO). For feature enhancement, the Multi-Grain Feature Extraction (MGFE) Structure was constructed to improve the multi-grain feature extraction capability of the proposed method.

In summary, the main contributions of this paper are as follows:The Joint Modal Alignment and Feature Enhancement (MAFE) network for visible-infrared person re-id is proposed to improve modal alignment and feature learning.Modal Align Operations (MAO), including Visible-Infrared Modal Data Augmentation (VIMDA) and Margin MMD-ID Loss, were introduced for modal alignment.The Multi-Grain Feature Extraction (MGFE) Structure was constructed to obtain multi-grain and discriminative features.Experiments on the SYSU-MM01 and the RegDB showed that our proposed method achieved the SOTA.

## 2. Related Works

### 2.1. Visible-Infrared Person Re-Identification

Visible-infrared person re-id is an extension of person re-id, which focuses on the problem of re-identification between visible and infrared images. Wu et al. [17] initiated the first attempt by proposing a deep zero-padding network to learn the common features under the two modalities and released the SYSU-MM01 dataset. Since then, visible-infrared person re-id has been widely noticed and studied by scholars. Most of the existing visible-infrared person re-id methods focus on how to reduce the modal difference between visible and infrared images and achieve modal alignment. These methods can be broadly classified into representation learning methods, metric learning methods, adversarial network-based methods and auxiliary image-based methods, depending on the pathway used to perform modal alignment. The representation learning methods [8,11] focus on how to improve the feature extraction process in both modals. Metric learning methods [16,18,19] focus on modifying the loss function during model training to optimize the feature metric space. Both representation learning methods and metric learning methods attempt to align modals during the feature-learning phase, but often have serious artificial design traces and bring complex network structures and calculations, which reduce the transferability of the model. Adversarial network-based methods [12,13,20] and auxiliary image-based methods attempt modality alignment at the pixel level of images. Compared to adversarial network-based methods, which require significant computational resources and introduce additional noise, auxiliary image-based methods aim to generate intermediate images between visible and infrared images. These intermediate images are then fed into the model for learning, resulting in a more simple and efficient approach. For example, Ye et al. [14] proposed a data augmentation that converted the original image into a random single-channel replica copied into three-channel image with 75% probability, or retained the original image with 25% probability, and then sent the augmented image into the model. This method achieved a great performance on visible-infrared person re-id, but it has some problems: the image conversion probability is high, meaning that the generated images have a high probability of resembling the infrared modality, but the retention probability is low, which may lead to the loss of useful color information for re-id. Conversely, when the conversion probability is low, the retention probability is high and the modal alignment may be poor. Therefore, for this method, it is necessary to explore the appropriate balance between image conversion and retention. Additionally, the image transformation operation can introduce more diverse training images, increasing the training difficulty of the Metric Loss in re-id and possibly resulting in poor model convergence.

### 2.2. Data Augmentation

Data augmentation methods, such as cropping, rotation, flipping and mix-up [21], use various data transformation operations to obtain multivariate data to train the model, thereby improving the effect of the model. In recent years, a number of data augmentation methods for visible-infrared person re-identification have also emerged. For example, one method [14] converted the visible image into random single-channel copy, then into a three-channel image. Another method, proposed in [22], involved converting the visible image directly into a grayscale image and using it as the input to the model; the method [23] involved performing data augmentation by randomly selecting one channel of the RGB image and setting the values of the other channels to zero. Overall, the aim of all these data augmentation methods is to reduce the sensitivity of the model to color information, reduce the difference between visible images and infrared images and achieve modal alignment. However, these methods are not tested for appropriate image conversion ratios and ignore some of the color information that contributes to the recognition results.

### 2.3. Feature Extraction

With the rapid development of deep learning, feature extraction based on convolutional neural networks and Transformer networks has gradually replaced traditional feature extraction methods. Due to its small model size and fast computation, the convolutional network is more suitable for pedestrian re-identification tasks compared to the Transformer feature extraction structure, which has high computational requirements. The representative of convolutional backbone networks is the Resnet [24] network, which was proposed by He et al. in 2015. Many methods have been developed to improve Resnet, such as ResNeXt [25], which introduces the multi-branch structure of the Inception [26] network into residual blocks to enhance the model’s multi-granularity information perception ability, and Resnest [27], which introduces attention mechanisms on this basis to obtain better feature extraction ability. In order to obtain more discriminating features upon the backbones, mining images for more granular information is a common means. For example, in object detection tasks, a common operation to enhance the detection effect of small objects is to continuously fuse shallow features with deep features; for multi-scale object detection, the paper [28] proposes to use multiple dilation rates of dilated convolution to make the model have multi-scale receptive fields and improve the model’s perception ability to different granularity information. This paper explores the multi-grain information in visible and infrared images from the perspective of feature extraction to improve the re-id performance.

## 3. Method

In this section, we first provide an overview of the overall approach of our method in Section 3.1. Then, in Section 3.2, we introduce the Modal Alignment Operations (MAO) used in this paper, which includes the Visible-Infrared Modal Data Augmentation (VIMDA) for image preprocessing and the Margin MMD-ID Loss for training. Next, in Section 3.3, we describe the Multi-Grain Feature Extraction (MGFE) Structure proposed in this work. Finally, in Section 3.4, we present all the loss functions that make up our model.

### 3.1. Overview

We used Resnest50 [27], pre-trained on ImageNet, as the backbone, as is shown in Figure 1. Resnest50 is a superior backbone network for feature extraction compared to Resnet50, and its multi-branch and multi-perceptual field structure can effectively extract multi-grain information. Additionally, following the AGW [1], we incorporated the Non-Local [29] attention mechanism into Layer 3 and Layer 4 of Resnest50 to enhance the network’s spatial relationship modeling capabilities. Specifically, the Non-Local attention was embedded after the last two Blocks in Layer 2 of Resnest50, with each Block embedding a Non-Local attention mechanism. The Non-Local attention was also embedded after the last three Blocks in Layer 3 of Resnest50, with each Block embedding the Non-Local attention mechanism. This can be seen in Figure 2.

Our proposed model is a dual-stream structure based on Resnest50, as shown in Figure 1. The initial Stem Layer, which includes the initial 7×7 convolutional layer, BN layer, ReLu layer and Max-pooling layer, is parameterized independently, while the other parts share weight parameters.

During the feature extraction stage, visible images are first input into the network, followed by the application of Visible-Infrared Modal Data Augmentation (VIMDA) and then the backbone network’s Stem. The Layer1–3 layers are used to extract preliminary feature maps. Next, the Multi-Grain Feature Extraction (MGFE) Structure is applied to enhance the multi-grain information components in the feature map. Finally, Layer 4 is used to extract features to generate the final visible feature map. The feature extraction process for infrared images does not involve VIMDA but is otherwise consistent with the feature extraction process for visible images. The feature extraction for visible and infrared images can be seen in Figure 1 and Equation (1).
(1)$$\begin{array}{l} F_{\mathrm{vis}}^{C\times H\times W}=\underset{{\mathrm{Layer}}_{4}}{Conv}\left(\mathrm{MGFE}\left(\underset{\mathrm{Stem},\;{\mathrm{Layer}}_{1–3}}{Conv}\left(\mathrm{VIMDA}\left(X_{\mathrm{vis}}\right)\right)\right)\right)\\ F_{\inf}^{C\times H\times W}=\underset{{\mathrm{Layer}}_{4}}{Conv}\left(\mathrm{MGFE}\left(\underset{\mathrm{Stem},\;{\mathrm{Layer}}_{1–3}}{Conv}\left(X_{\inf}\right)\right)\right) \end{array}$$
where $X_{\mathrm{vis}}$ and $X_{\inf}$, respectively, represent the visible image and the infrared image; $F_{\mathrm{vis}}^{C\times H\times W}$ and $F_{\inf}^{C\times H\times W}$ represent the feature map; $\underset{\mathrm{Stem},\;{\mathrm{Layer}}_{1–3}}{Conv}$ and $\underset{{\mathrm{Layer}}_{4}}{Conv}$, respectively, represent the convolutional operations in the proposed method’s Stem Layer from Layer 1 to Layer 3 and Layer 4 as shown in Figure 1; and VIMDA and MGFE, respectively, represent the Visible-Infrared Modal Data Augmentation and the Multi-Grain Feature Extraction Structure.

During the model training phase, firstly, all feature maps are subjected to GeM [30] pooling to output feature vectors. Then, the Metric Loss (including Hc-Tri Loss [19] and the Enhanced WRT Loss [14]) and the Margin MMD-ID Loss [16] between the visible and infrared feature vectors are calculated. After that, the feature vectors are normalized using Batch Normalization (BN) and mapped using fully connected layers to compute the ID Loss (Cross-Entropy Loss). During the model inference phase, BN-normalized feature vectors are used for inference, as is shown in Figure 1.

### 3.2. Modal Alignment Operations (MAO)

In the proposed method, we first propose a new data augmentation method called Visible-Infrared Modal Data Augmentation (VIMDA) to reduce the modal difference, which incorporates a grayscale conversion operation into the previous CAJ method [14], since grayscale images have a similar appearance to the single-channel image and are the integration of all RGB color channels. The specific implementation process of the VIMDA is shown in Figure 3a. When given an RGB image as input, the original image may be randomly modified or replaced based on fixed probabilities. Specifically, the original image has a probability of “a%” to be preserved, a probability of “b%” to be replaced with a grayscale image or a probability of “c%” to be replaced with a new three-channel image generated by replicating a random single channel. The probability ratio of a, b and c is determined through well-designed experiments, and the sum of all probabilities is equal to one, as shown in Equation (2).

In general, the difference between VIMDA and the data augmentation in the CAJ method [14] is that VIMDA introduces an additional grayscale conversion operation, and the probability ratios in VIMDA are carefully designed through experiments, as shown in Figure 3a,b.
(2)a%+b%+c%×3=100%

Due to the use of VIMDA, the input data of the visible branch in the proposed model have changed from a single RGB image to a diverse and complex input. During the training process, we found that such an increase in input for the visible branch made the convergence of the Metric Loss more difficult and even occasionally caused non-convergence. Therefore, this paper introduces the Margin MMD-ID to constrain the distance between the output distributions of the visible branch and the infrared branch, achieving modal alignment and helping the model converge, as is shown in Figure 1. The formula for the Margin MMD-ID loss is shown in Equation (3).
(3)$$\begin{array}{c} MMD^{2}\left(P^{c},Q^{c}\right)=\frac{1}{n^{2}}\sum_{i=1}^{n}\sum_{i^{\prime }=1}^{n}k\left(x_{v}^{i},x_{v}^{i^{\prime }}\right)+\frac{1}{m^{2}}\sum_{j=1}^{m}\sum_{j^{\prime }=1}^{m}k\left(x_{t}^{j},x_{t}^{j^{\prime }}\right)-\\ \frac{2}{nm}\sum_{i=1}^{n}\sum_{j=1}^{m}k\left(x_{v}^{i},x_{t}^{j}\right),\;k\left(u,v\right)=e^{\frac{-{\left\|u-v\right\|}^{2}}{\sigma }}\\ L_{\mathrm{Margin}\;\mathrm{MMD}\text{-}\mathrm{ID}}=\left\{\begin{array}{c} MMD^{2}\left(P^{c},Q^{c}\right),\mathrm{if}\;MMD^{2}\left(P^{c},Q^{c}\right)>\rho_{1}\\ 0,\mathrm{otherwise} \end{array}\right. \end{array}$$
where $MMD^{2}\left(P^{c},Q^{c}\right)$ denotes the Maximum Mean Discrepancy for person c, $P^{c}$ and $Q^{c}$, respectively, represent the probability distribution of visible images and infrared images of person c, n denotes the number of visible images of person c, m denotes the number of infrared images of person c, $x_{v}^{i}$ and $x_{t}^{j}$ denote visible features and infrared features of person c, $k\left(u,v\right)$ denotes the Gaussian kernel function, $\rho_{1}$ denotes the margin hyper-parameter, and $L_{\mathrm{Margin}\text{-}\mathrm{MMD}\text{-}\mathrm{ID}}$ denotes the Margin MMD-ID Loss.

### 3.3. Multi-Grain Feature Extraction (MGFE) Structure

In order to improve the representation learning ability of the model, this paper proposes the Multi-Grain Feature Extraction (MGFE) Structure and embeds it between Layer 3 and Layer 4 of the backbone. MGFE contains two parts: the Dilated Encoder [28] module (DE) and the Cross-Layer Feature Fusion (CLFF), as shown in Figure 4.

The DE module contains two parts, the Projector and the Residual Blocks, as shown in Figure 5. In this, the Residual Blocks are obtained by concatenating four residual blocks, each of which contains a dilated convolution with a different dilated ratio(2, 4, 6, 8). The use of cascading dilated convolutions with different dilated ratios allows the model to obtain a more multi-scale perceptual field and improves the ability of the model to output features containing multi-grain information. In MGFE, the DE module is embedded behind Layer 3 in the form of residual connection, as is shown in Figure 4.

The CLFF fuses the output of Layer 1 with the residual output of the DE module. The specific feature fusion is performed by down sampling the output of Layer 1 with a 1*1 convolutional layer with a stride size of 4, then concatenating it with the residual output of the DE module in the channel dimension and finally dimensionally transforming it with a 1*1 convolutional layer before feeding it into Layer 4 for further information extraction and integration.

### 3.4. Loss Function

In addition to using the Margin MMD-ID Loss to constrain the distance between the visible features distribution and the infrared features distribution after Gem pooling, the Metric Loss, including Hc-Tri Loss [19] and the Enhanced WRT Loss [14] as is shown in Equations (4) and (5), is also applied to all pooled features to further optimize the relative distances between features and feature centers in the metric space. Simultaneously, the Cross-Entropy Loss is calculated as the ID Loss for the features following the fully connected layer. Finally, the total loss function during model training phase is shown in Figure 6 and Equation (6).
(4)$$\begin{array}{c} L_{\mathrm{Hc}\text{\_}\mathrm{Tri}}={\sum_{i=1}^{P}\left[\rho_{2}+∥c_{v}^{i}-c_{t}^{i}{∥}_{2}-\underset{n\in \left\{v,t\right\},j\neq i}{\min}∥c_{v}^{i}-c_{n}^{j}{∥}_{2}\right]}_{+}\\ +{\sum_{i=1}^{P}\left[\rho_{2}+∥c_{t}^{i}-c_{v}^{i}{∥}_{2}-\underset{n\in \left\{v,t\right\},j\neq i}{\min}∥c_{t}^{i}-c_{n}^{j}{∥}_{2}\right]}_{+},\\ c_{v}^{i}=\frac{1}{K}\sum_{k=1}^{K}v_{k}^{i},c_{t}^{i}=\frac{1}{K}\sum_{k=1}^{K}t_{k}^{i} \end{array}$$
where $c_{v}^{i}$ and $c_{t}^{i}$, respectively, represent the feature centroid of the visible images and the infrared images of person i and $c_{n}^{j}$ represents the centroid of all visible and infrared image features of a specific person, excluding person i. P represents the number of persons, K represents the number of visible images or infrared images per person during training, $v^{i}$ and $t^{i}$ represent the visible features and infrared features of person i and $\rho_{2}$ represents the margin hyper-parameter of the Hc-Tri Loss.
(5)$$\begin{array}{c} L_{\mathrm{Enhanced}\;\mathrm{WRT}}=\frac{1}{N}\sum_{i=1}^{N}\log\left(1+\exp\left(\phi \left[\sum_{ij}w_{ij}^{p}d_{ij}^{p}-\sum_{ik}w_{ik}^{n}d_{ik}^{n}\right]\right)\right),\\ w_{ij}^{p}=\frac{\exp\left(d_{ij}^{p}\right)}{\sum_{d_{ij}^{p}\in P_{i}}\exp\left(d_{ij}^{p}\right)},w_{ik}^{n}=\frac{\exp\left(-d_{ik}^{n}\right)}{\sum_{d_{ik}^{n}\in N_{i}}\exp\left(-d_{ik}^{n}\right)},\\ \phi \left[\mu_{i}\right]=\left\{\begin{array}{c} \mu_{i}^{2},\mu_{i}^{2}>0,\\ -\mu_{i}^{2},\mu_{i}^{2}<0. \end{array}\right. \end{array}$$
where i represents any sample, j represents any positive sample of i and k represents any negative sample of i. $P_{i}$ represents the set of positive samples of i, $N_{i}$ represents the set of negative samples of i, $d_{ij}^{p}$ represents the Euclidean distance between i and any positive sample j, $d_{ik}^{n}$ represents the Euclidean distance between i and any negative sample k, N represents the number of samples in a batch, $\sum_{ij}w_{ij}^{p}d_{ij}^{p}$ represents the weighted sum of distances to positive samples and $\sum_{ik}w_{ik}^{n}d_{ik}^{n}$ represents the weighted sum of distances to negative samples.
(6)$$\begin{array}{cc} L_{\mathrm{Total}} & =L_{\mathrm{Enhanced}\;\mathrm{WRT}}+L_{C\mathrm{ross}\text{-}Entropy}+\\ & \lambda_{1}L_{\mathrm{Margin}\;\mathrm{MMD}\text{-}\mathrm{ID}}+\lambda_{2}L_{\mathrm{Hc}\text{\_}\mathrm{Tri}} \end{array}$$
where $L_{\mathrm{Total}}$ denotes the total loss function in the proposed model while $\lambda_{1}$ and $\lambda_{2}$ denote the weight hyper-parameters of the Margin MMD-ID Loss and the Hc-Tri Loss.

## 4. Experiments

### 4.1. Datasets and Evaluation Metric

#### 4.1.1. Datasets

Our proposed method was trained and evaluated on two visible-infrared person re-id datasets, SYSU-MM01 [17] and RegDB [31].

**SYSU-MM01** was obtained from 2 indoor visible cameras, 2 outdoor visible cameras and 2 near infrared cameras. For the training phase, 22,258 visible images and 11,909 infrared images from 395 persons were used. Following [17], the testing phase involved two modes: All Search mode and Indoor Search mode. In All Search mode, we utilized 3803 infrared images of 96 pedestrians from the test set as queries and selected 301 images at random from all visible images in the test set as the gallery for recognition. In Indoor Search mode, we used 3803 infrared images of the same 96 pedestrians from the test set as queries and selected 112 images at random from the indoor visible camera in the test set as the gallery for recognition.

**The RegDB** dataset comprised 412 pedestrians, captured using a visible light camera and a thermal imaging camera. Each pedestrian had 10 visible images and 10 infrared images. The dataset was divided into a training set and a test set, with all images of 206 pedestrians used for training and the remaining 206 pedestrians used for testing.

#### 4.1.2. Evaluation Metric

This paper evaluates the performance of the proposed model using the Cumulative Match Characteristic (CMC) and mean Average Precision (mAP) metrics.

### 4.2. Experimental Settings

#### 4.2.1. Basic Settings

The experiments were carried out with Pytorch on a single NVIDIA TITAN V GPU. During training, in addition to using VIMDA for visible images, cropping, random erasing and random horizontal flipping were performed on all visible light and infrared images. During model inference, no data augmentation was performed on the input image. In VIMDA, the probability of retaining the visible image was 40%, the probability of converting to grayscale was 30% and the probability of converting to a new three-channel image copied from any single channel was 30%. The stride of the down-sampling convolution layer between Layer 3 and Layer 4 was set to 1. The margin hyper-parameters in the Margin MMD-ID Loss and the Hc-Tri Loss were 1.7 and 4, and the weight hyper-parameters of the Margin MMD-ID Loss and the Hc-Tri Loss in the total loss function were 0.12 and 0.05. We used the stochastic gradient descent method (SGD) and the dynamic learning rate to update the model gradient, as shown in (7). The number of iterative epochs during training was 100; each training batch contained 8 people, and each person had 4 visible images and 4 infrared images.
(7)$$lr\left(epoch\right)=\left\{\begin{array}{c} 0.1\times \frac{epoch+1}{10},epoch<10\\ 0.1,10\leq epoch<20\\ 0.01,20\leq epoch<50\\ 0.001,epoch\geq 50 \end{array}\right.$$

#### 4.2.2. Test Settings

For the SYSU-MM01 dataset, All Search mode and Indoor Search mode were adopted to test the performance of the proposed method. Specifically, we conducted 10 random tests and computed the average CMC and mAP scores across the 10 runs as the overall performance of the model. For the RegDB dataset, the test set consisted of two modes: visible image re-identification infrared image (Visible to Infrared) and infrared image re-identification visible image (Infrared to Visible). As the RegDB dataset has a small volume, we divided both the training and test sets into 10 subsets and performed a 10-fold cross-validation to obtain the final CMC and mAP scores of the model.

### 4.3. Comparison with Existing Methods

We compared our proposed method on two datasets, SYSU-MM01 and RegDB, with the current state-of-the-art methods, including Hi-CMD [32], X-Modal [33], DDAG [34], HAT [35], NFS [6], MSO [18], CM-NAS [7], MCLNet [20], SMCL [9], CAJ [14], MPANet [10], FMCNet [36] and MAUM [37]. The comparison results are presented in Table 1, where the data in bold indicates the highest value of all methods.

As shown in the table, on the All Search test mode of the SYSU-MM01 dataset, our proposed method achieved Rank1 and mAP of 74.85% and 71.7%, respectively, which are 3.17% and 2.91% higher than the best MAUM model, respectively. On the Indoor Search test mode of the SYSU-MM01 dataset, our proposed method achieved Rank1 and mAP of 80.85% and 84.06%, respectively, which are 3.08% and 2.21% higher than the best MAUM model, respectively. On the Visible to Infrared test mode of the RegDB dataset, our proposed method achieved Rank1 and mAP values of 91.17% and 81.81%, respectively, with a 3.3% increase in Rank1 compared to the MAUM model, but a decrease in mAP. On the Infrared to Visible test mode of the RegDB dataset, our proposed method achieved Rank1 and mAP values of 90.05% and 80.27%, respectively, with a 3.55% increase in Rank1 compared to the MAUM model, but a 4.07% decrease in mAP. Upon comparing the results, it is evident that our proposed method MAFE outperforms other existing methods on the SYSU-MM01 dataset; on the RegDB dataset, MAFE achieves a significantly higher Rank1 value than other existing methods, although the mAP value falls short of the MAMU method. These results serve as a testament to the effectiveness and superiority of the MAFE.

### 4.4. Ablation Study

#### 4.4.1. Ablation Experiments of Each Component

We conducted ablation experiments on the All Search mode of SYSU-MM01 to evaluate the impact of the MAO (including VIMDA and Margin MMD-ID Loss) and the MGFE (including CLFF and DE) on re-id performance. The results are presented in Table 2. The findings indicate that each component in our proposed method effectively improves the recognition performance. With the loading of all components, the Rank1 and mAP significantly increased by 6.15% and 6%, respectively, compared to Experiment 1. Our experimental results demonstrate the effectiveness of each component in our proposed method for visible-infrared person re-identification.

#### 4.4.2. Analysis of the probability ratio in VIMDA

In VIMDA, if too much of the original visible image is retained, the modal alignment will not be achieved. Otherwise, some color information that may be beneficial to person re-identification is lost. Therefore, this paper seeks the optimal probability ratio by fine-tuning the transformation probabilities in VIMDA. The experiment was tested on the All-Search mode of the SYSU-MM01 dataset, and the experimental content as well as the experimental results are shown in Figure 7. As can be seen from the figure, the re-id performance of the model gradually decreased as the proportion of visible images retained increased, and the model reached an optimal re-id performance of Rank1–74.85% and mAP–71.7% when the probability ratio between retaining visible images, converting to grayscale images or converting to random single-channel replication into three-channel images was 4:3:3.

#### 4.4.3. Ablation Experiments of Cross-Layer Feature Fusion (CLFF)

In the All Search mode of the SYSU-MM01 dataset, experiments were conducted on the influence of different layer feature fusion in the CLFF of MGFE. The experimental content and results are shown in Table 3, where the numbers denote the output of the corresponding layer and the R-O denotes the residual output of the DE module. The table clearly indicates that the cross-layer fusion of the output from Layer 1 and the residual output from the DE module resulted in the best re-id performance.

#### 4.4.4. Analysis of Different Backbones

We built two-stream networks using different backbones (Resnet50, Res2Net50 [38] and Resnest50) to test the effectiveness and robustness of the proposed method (MAO and MGFE) in this paper, as shown in Table 4.

The results presented in Table 4 provide strong evidence that the proposed method (MAO and MGFE) significantly improves visible-infrared re-id performance across various backbone networks. In particular, the Resnest50 backbone with MAO and MGFE added yielded the highest re-id accuracy, with Rank1–74.85% and mAP–71.7%. These findings effectively demonstrate the effectiveness and robustness of the method proposed in this paper.

### 4.5. Visualization and Analysis

#### 4.5.1. Heat Map Analysis of MGFE

We used Grad-Cam [39] to visually analyze the Layer 4 features extracted from the network model with or without MGFE. As can be seen from Figure 8, the embedding of MGFE, on the one hand, enabled the model to better focus on the person itself and distinguish the background; on the other hand, it enabled the model to focus on more identifiable features of different scales in different parts of the person image. The visual heat map clearly illustrates that MGFE significantly enhanced the model’s feature-learning capability and improved the performance of visible-infrared person re-id.

#### 4.5.2. Output of Visible-Infrared Person Re-ID Results

In the SYSU-MM01 and RegDB test sets, we selected several infrared images of individuals as the Query and used the method proposed in the paper to match the visible person images in the corresponding Gallery set. We then visualized the results, as shown in Figure 9 and Figure 10. The number displayed above the output of the results represents the similarity ranking with the Query. The lower the number, the higher the similarity between the images. The green box indicates the correctly identified sample, while the red box indicates the incorrectly identified sample.

The re-identification results of the SYSU-MM01 dataset in Figure 9 demonstrates that the proposed method can accurately retrieve all correct samples for simple cases, while for more challenging persons, it can still identify them up to Rank 5. For the RegDB dataset, as is shown in Figure 10, our model can effectively retrieve all easy samples, but due to the poor quality of infrared images in the RegDB dataset and the lack of detailed information, it is difficult for the model to accurately distinguish and recognize some of the easily confused samples.

## 5. Discussion

### 5.1. Next Challenges and Future Perspective

For visible-infrared person re-id, how to improve the model to extract common information from images captured under two modalities and generate more discriminative features is a crucial challenge that restricts the improvement of accuracy. From the results in our paper, we can find that it is worth studying how to improve the recognition effect of the model on cross-modal datasets lacking a large amount of detailed information, such as the RegDB dataset. Additionally, enabling the convolutional model to model long-distance relationships like the Transformer is also a challenging task.

### 5.2. Replicability of the Method

The details and parameters of the proposed method have been presented in Section 3. Method and Section 4.2. Experimental Settings. By following these method details and parameters, the results of this paper can be fully replicated.

## 6. Conclusions

This paper proposes the Joint Modal Alignment and Feature Enhancement (MAFE) network for visible-infrared person re-id, whose main focus is on modal alignment and feature enhancement. First, it proposes VIMDA and introduces Margin MMD-ID Loss for modal alignment. Then, the Multi-Grain Feature Extraction (MGFE) Structure is proposed and embedded to extract features that contain multi-grain information and are more discriminative. A large number of experiments were carried out on the method of this paper with the SYSU-MM01 dataset and the RegDB dataset. The experimental results fully demonstrate the effectiveness of the proposed method.

## Figures and Tables

**Figure 1 sensors-23-04988-f001:**
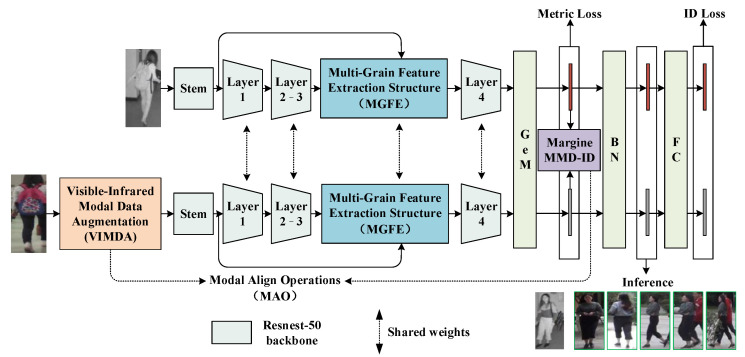
Overall architecture of the proposed method.

**Figure 2 sensors-23-04988-f002:**
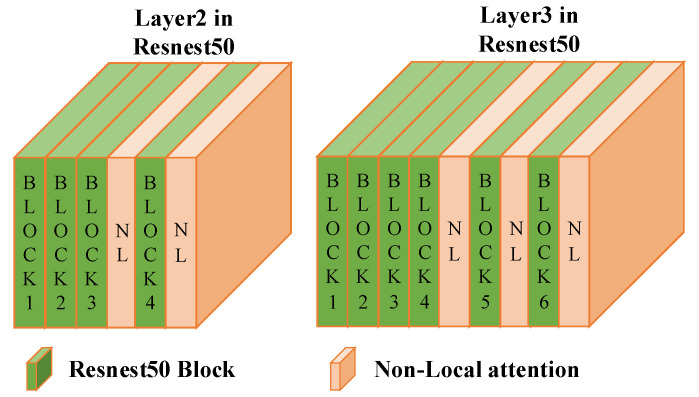
Non-Local attention in the proposed method.

**Figure 3 sensors-23-04988-f003:**
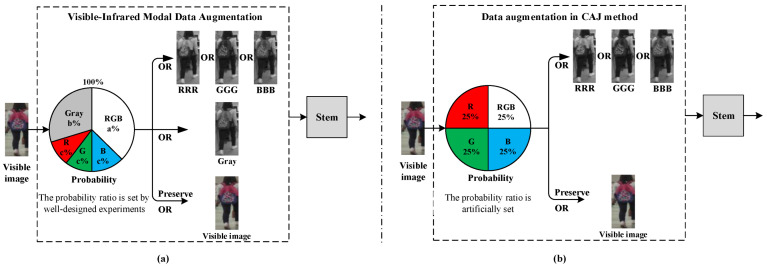
The Visible-Infrared Modal Data Augmentation (VIMDA) and the data augmentation in the CAJ method [14]. (**a**) VIMDA. (**b**) Data augmentation in CAJ method.

**Figure 4 sensors-23-04988-f004:**
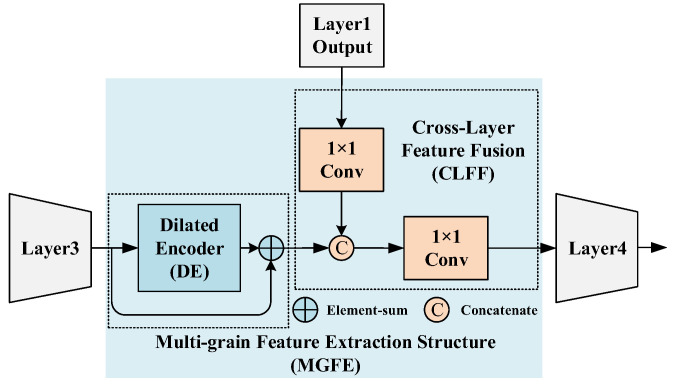
Multi-Grain Feature Extraction Structure (MGFE).

**Figure 5 sensors-23-04988-f005:**
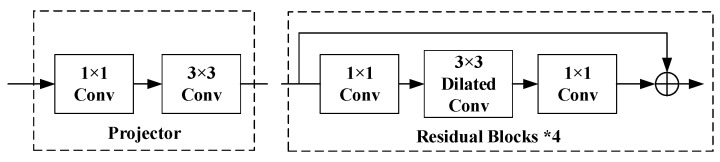
Dilated Encoder (DE) module.

**Figure 6 sensors-23-04988-f006:**
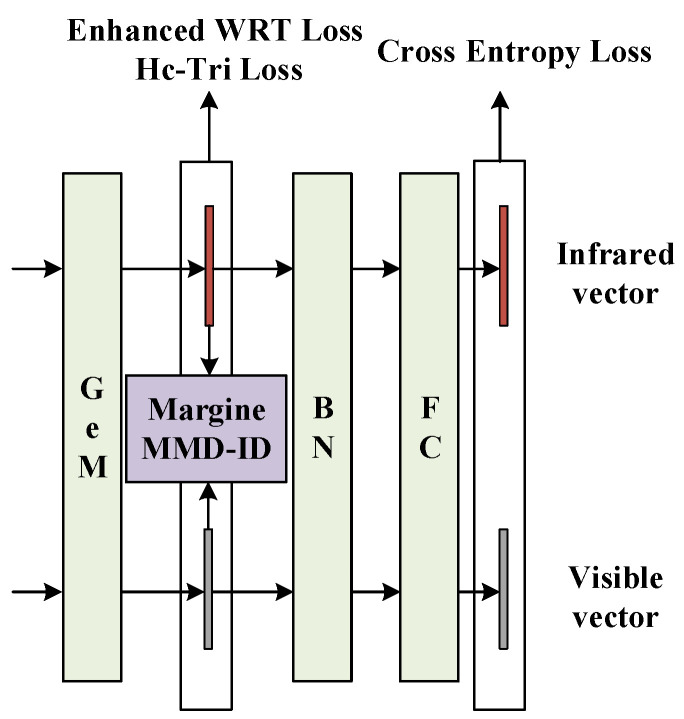
Overall loss function.

**Figure 7 sensors-23-04988-f007:**
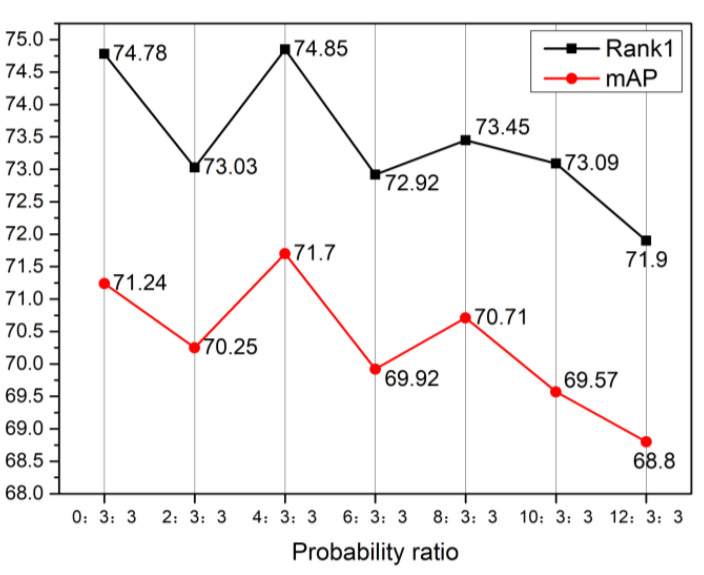
Experimental graph for the probability ratio in VIMDA; the abscissa in the graph represents the proportion of probability between preserving the original visible image, converting to grayscale image or converting to random single-channel replication into three-channel images.

**Figure 8 sensors-23-04988-f008:**
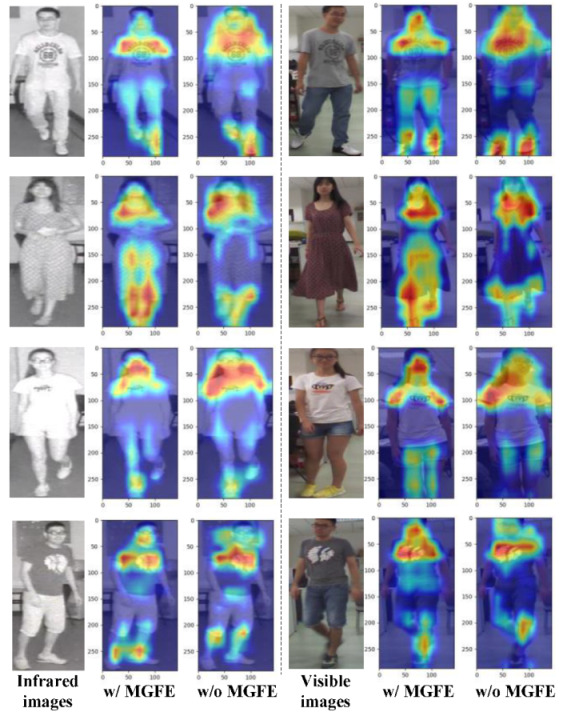
We used Grad-Cam to visually analyze the Layer 4 features extracted from the model with or without MGFE. Three pictures are a group, the left picture in each group is an infrared image or visible image, the middle picture is a heat map with MGFE and the right picture is a heat map without MGFE.

**Figure 9 sensors-23-04988-f009:**
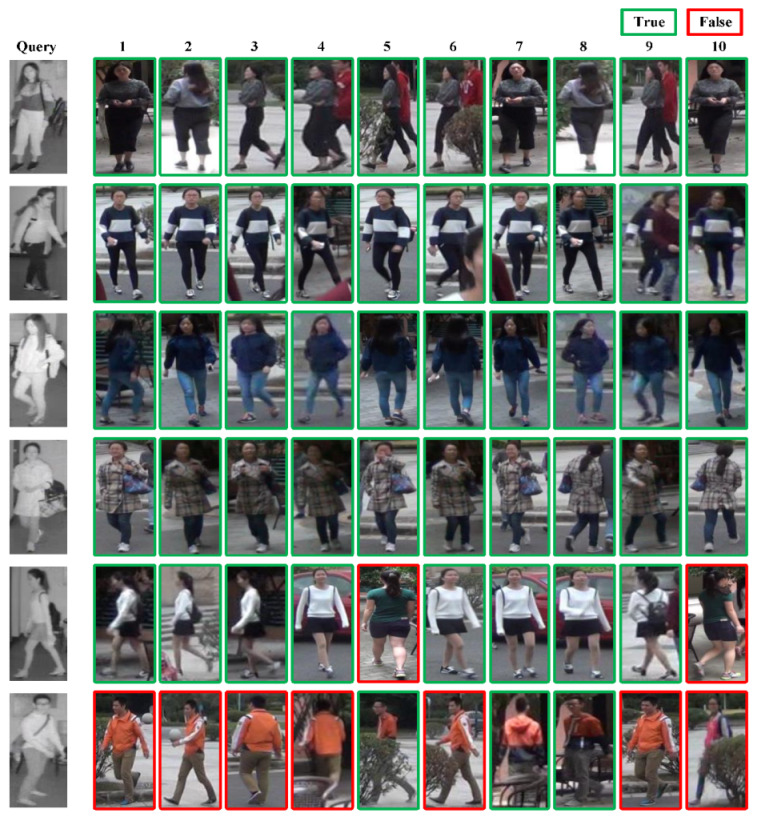
Results output on the SYSU-MM01 dataset.

**Figure 10 sensors-23-04988-f010:**
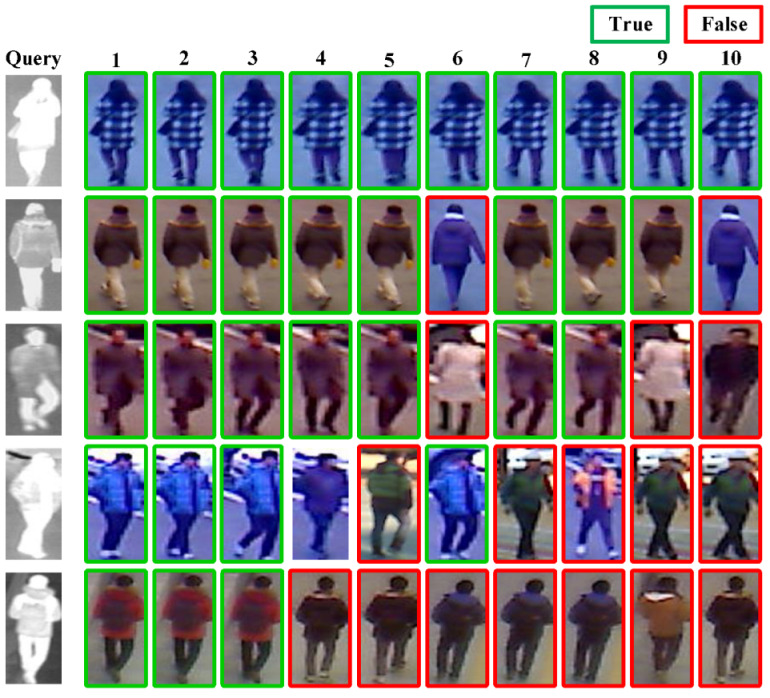
Results output on the RegDB dataset.

**Table 1 sensors-23-04988-t001:** Performance comparison with existing methods.

Method	Venue	SYSU-MM01				RegDB			
		All Search		Indoor Search		Visible to Infrared		Infrared to Visible	
		Rank1	mAP	Rank1	mAP	Rank1	mAP	Rank1	mAP
Hi-CMD [32]	CVPR2020	34.9	35.9	-	-	70.93	66.04	-	-
X-Modal [33]	AAAI2020	49.9	50.7	-	-	62.21	60.18	-	-
DDAG [34]	ECCV2020	54.75	53.02	61.02	67.98	69.34	63.46	68.06	61.8
HAT [35]	TIFS2020	55.29	53.89	62.1	69.37	71.83	67.56	70.02	66.3
NFS [6]	CVPR2021	56.91	55.45	62.76	69.79	80.54	72.1	77.95	69.79
MSO [18]	ACMMM2021	58.7	56.42	63.09	70.31	73.6	66.9	74.6	67.5
CM-NAS [7]	ICCV2021	61.99	60.02	67.01	72.95	84.54	80.32	82.57	78.31
MCLNet [20]	ICCV2021	65.4	61.98	72.56	76.58	80.31	73.07	75.93	69.49
SMCL [9]	ICCV2021	67.39	61.78	68.84	75.56	83.93	79.83	83.05	78.57
CAJ [14]	ICCV2021	69.88	66.89	76.26	80.37	85.03	79.14	84.75	77.82
MPANet [10]	CVPR2021	70.58	68.24	76.74	80.95	82.8	80.7	83.7	80.9
FMCNet [36]	CVPR2022	66.34	62.51	68.15	74.09	89.12	84.43	88.38	83.86
MAUM [37]	CVPR2022	71.68	68.79	76.97	81.94	87.87	**85.09**	86.95	**84.34**
**MAFE**	**ours**	**74.85**	**71.7**	**80.85**	**84.06**	**91.17**	81.81	**90.05**	80.27

**Table 2 sensors-23-04988-t002:** Ablation experiments of each component.

No.	MAO		MGFE		SYSU-MM01 (All Search)	
	VIMDA	Margin MMD-ID Loss	CLFF	DE	Rank1	mAP
1	-	-	-	-	68.7	65.7
2	**√**	-	-	-	70.57	68.04
3	**√**	**√**	-	-	71.1	68.03
4	**√**	**√**	-	**√**	72.02	68.84
5	**√**	**√**	**√**	-	73.23	70.47
6	**√**	**√**	**√**	**√**	**74.85**	**71.7**

**Table 3 sensors-23-04988-t003:** Experiments for Cross-Layer Feature Fusion.

Feature Fusion	SYSU-MM01 (All Search)	
	Rank1	mAP
0→2	69.63	66.34
1→2	68.74	65.41
0,1→2	67.73	64.31
0→R_O	73.66	70.93
1→R_O	**74.85**	**71.7**
2→R_O	72.52	69.59
0,1→R_O	73.64	70.04
0,2→R_O	74.8	71.39
1,2→R_O	72.56	68.41
0,1,2→R_O	73.48	70.01

**Table 4 sensors-23-04988-t004:** Experiments for different backbones.

Method	SYSU-MM01 (All Search)	
	Rank1	mAP
Resnet50	64.93	62.26
Resnet50_with_MAO and MGFE	69.27	65.42
Res2Net50	62.02	60.02
Res2Net50_with_MAO and MGFE	69.21	66.52
Resnest50	68.7	65.7
Resnest50_with_MAO and MGFE (proposed method)	**74.85**	**71.7**

## Data Availability

The data that support the findings of this study are available online. These datasets were derived from the following public resources: [SYSU-MM01, RegDB].

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
