# Peer review of "Joint Modal Alignment and Feature Enhancement for Visible-Infrared Person Re-Identification"

_sensors, 2023, doi:10.3390/s23114988_

Round 1

Reviewer 1 Report

Regarding the article sensors-2387180 titled Joint Modal Alignment and Feature Enhancement for Visible-Infrared Person Re-identification and they have clearly demonstrated the advantages of their approach in comparison to others in the literature.

Authors use a readable English, and some parts are easy to understand. In the same way, they state the justification and main problem.

Furthermore, Authors expose the justification, and I could find the explanation of the main problem in a clearly way.

Then, this paper is based on 34 references from 2016 to 2023 (7 years of research) , and the authors  consider recent references. Only 80% are  from the last five years (from 2018 to 2023) In addition, the references are relevant to the topic they are introducing.

Originality Report shows that this article has a similarity index of 13%, which can be considered as original work. This similarity report is attached to this review.

Please consider the following remarks to improve your article (in some cases, P refers to Page or Pages and L is the Line or Lines where you can find these remarks):

·         Section References is complete, since the most of references  (80%) can be considered as of recent works.

·         The similarity Index of this work is 13%, so it can be considered as original work .

·         The problem and justification are well described.

·         The comparison between state-of-the-art algorithm is not complete at all.

·         Well distribution of the elements to be described or analyzed

·         There is  not enought experimentation or comparison of the results that demonstrate the novelty of the project.

·         Results improperly exposed

·         Authors use  comparative table of the characteristics of the related work, in addition use some plots that help the experimental results.

·         Equations  are not well described or defined

·         Authors include general algorithm of the proposed work.

·         So many details were omitted in the methodology that it is difficult to estimate the contribution of the article.

So, I suggest modifying, if it is the case, for the publication in the journal Sensors (1424-8220), since the paper by itself have a great potential to publish.

Regards.

Authors use a readable English, and some parts are easy to understand. In the same way, they state the justification and main problem.

Furthermore, Authors expose the justification, and I could find the explanation of the main problem in a clearly way.

Then, this paper is based on 34 references from 2016 to 2023 (7 years of research) , and the authors  consider recent references. Only 80% are from the last five years (from 2018 to 2023) In addition, the references are relevant to the topic they are introducing.

Author Response

Response to Reviewer

Dear Reviewer:

We feel great thanks for your professional review work on our article. As you are concerned, there are several problems that need to be addressed. According to your nice suggestions, we have made extensive corrections to our previous draft, the detailed corrections are listed below.

Point 1: The comparison between state-of-the-art algorithm is not complete at all.

Response 1: Dear reviewer, regarding your comments, we have added several methods from the year 2020 and 2022 to the table of comparative experiments. We have also provided appropriate descriptions and analyses of the comparative results in the "4.3 Comparison with Existing Methods" section.

Point 2: There is not enought experimentation or comparison of the results that demonstrate the novelty of the project.

Response 2: Dear reviewer, regarding your comments, we have made modifications to the content of Figure 3 by adding a comparative chart to highlight the differences between our proposed Visible-Infrared Modal Data Augmentation (VIMDA) method and other approaches, as well as to emphasize the novelty and innovation of our method. Furthermore, we have provided an introduction and summary of the innovation of VIMDA in the "3.2. Modal Alignment Operations (MAO)" section.

Point 3: Results improperly exposed.

Response 3: Dear reviewer, regarding your comments, we have made modifications to the output graph of SYSU-MM01 and added the output graph of the model on the RegDB dataset. Additionally, we have provided appropriate analyses and explanations for all output graphs in the "4.5.2 Output of Visible-Infrared Person Re-ID Results" section.

Point 4: Equations are not well described or defined.

Response 4: Dear reviewer, in response to your comment, we have added the missing formulas for the Hc-Tri Loss and Enhanced WRT Loss, and provided detailed descriptions for each formula.

Point 5: So many details were omitted in the methodology that it is difficult to estimate the contribution of the article.

Response 5: Dear reviewer, in response to your comment, we have added the embedding position of the Non-Local attention module in our model and provided an illustration to explain it. Other details of our proposed method and some model parameter settings are described in detail in the "4.2. Experimental Settings" section.

Reviewer 2 Report

This interesting work is based on Joint Modal Alignment and Feature Enhancement for Visible-2 Infrared Person Re-identification. The research and the manuscript are conducted well, howerver I suggest you to follow the list below for further fixes: - I suggest you to incude a discussion chapter as well as to implement the conclusion chapter with other consideration; - Future perspective, next challenges and the replicability of the method can be included into the manuscript; - At the same time, could be a proper solution the implementation of international bibliographic references. Thank you

Author Response

Response to Reviewer

Dear Reviewer:

We feel great thanks for your professional review work on our article. As you are concerned, there are several problems that need to be addressed. According to your nice suggestions, we have made extensive corrections to our previous draft, the detailed corrections are listed below.

Point : The research and the manuscript are conducted well, howerver I suggest you to follow the list below for further fixes: - I suggest you to incude a discussion chapter as well as to implement the conclusion chapter with other consideration; - Future perspective, next challenges and the replicability of the method can be included into the manuscript; - At the same time, could be a proper solution the implementation of international bibliographic references.

Response: Dear reviewer, following your suggestion, we have added a chapter titled "5 Discussion" to provide appropriate analyses of the challenges we face and future prospects.

Apart from the above revisions, this paper also includes the following changes based on suggestions from other reviewers:

1 We have added several methods from the year 2020 and 2022 to the table of comparative experiments. We have also provided appropriate descriptions and analyses of the comparative results in the "4.3 Comparison with Existing Methods" section.

2 We have made modifications to the content of Figure 3 by adding a comparative chart to highlight the differences between our proposed Visible-Infrared Modal Data Augmentation (VIMDA) method and other approaches, as well as to emphasize the novelty and innovation of our method. Furthermore, we have provided an introduction and summary of the innovation of VIMDA in the "3.2. Modal Alignment Operations (MAO)" section.

3 We have made modifications to the output graph of SYSU-MM01 and added the output graph of the model on the RegDB dataset. Additionally, we have provided appropriate analyses and explanations for all output graphs in the "4.5.2 Output of Visible-Infrared Person Re-ID Results" section.

4 We have added the missing formulas for the Hc-Tri Loss and Enhanced WRT Loss, and provided detailed descriptions for each formula.

5 We have added the embedding position of the Non-Local attention module in our model and provided an illustration to explain it. Other details of our proposed method and some model parameter settings are described in detail in the "4.2. Experimental Settings" section.

Reviewer 3 Report

Person reidentification is a technique for retrieving and matching specific pedestrian images across cameras and is crucial for intelligent surveillance. Visible infrared person reidentification is a subtask of person re-id that aims to facilitate cross-modal retrieval and matching between visible and infrared person images. This technology has gained significant attention in the field of intelligent analysis for infrared cameras, which are widely deployed in cities to enhance citizen security. Given its capacity to accurately and efficiently recognize individuals in both day and night settings, visible infrared person re-id has the potential to significantly enhance public safety. This paper proposes Joint Modal Alignment and Feature Enhancement (MAFE) network for visible-infrared person re-id. For better modal alignment, the grey-scale conversion operation for visible images was introduced together with the data augmentation method, which is named Visible-infrared Modal Data Augmentation (VIMDA); at the same time, the probability ratio of images conversion and retention in VIMDA were tested to obtain the best performance. The authors carefully analyze all the aspects related to the technique: data augmentation, feature extraction with pertinent bibliographic references as support. The arguments supporting the research idea are supported mathematically. Analysis models are created for which the entry conditions are established. The authors propose a modal alignment operations method for image analysis. In order to improve the representation learning ability of the model, this paper proposes a multi-grain feature extraction. The simulation data is later verified on 412 people. Statistical analysis is complex.  In summary, the main contributions of this paper are as follows: - Joint Modal Alignment and Feature Enhancement (MAFE) network for visible-infrared person re-id is proposed to improve modal alignment and feature learning; Modal Align Operations (MAO), including Visible-Infrared Modal Data Augmentation (VIMDA) and Margin MMD-ID Loss, is introduced for modal alignment -The Multi-Grain Feature Extraction (MGFE) Structure is constructed to obtain multigrain and discriminative features. Experiments on the SYSU-MM01 and the RegDB show that our proposed method achieved the SOTA. 

Author Response

Response to Reviewer

Dear Reviewer:

We feel great thanks for your professional review work on our article. Based on the suggestions of other reviewers, we have made some revisions to the article. The changes are as follows:

1 We have added several methods from the year 2020 and 2022 to the table of comparative experiments. We have also provided appropriate descriptions and analyses of the comparative results in the "4.3 Comparison with Existing Methods" section.

2 We have made modifications to the content of Figure 3 by adding a comparative chart to highlight the differences between our proposed Visible-Infrared Modal Data Augmentation (VIMDA) method and other approaches, as well as to emphasize the novelty and innovation of our method. Furthermore, we have provided an introduction and summary of the innovation of VIMDA in the "3.2. Modal Alignment Operations (MAO)" section.

3 We have made modifications to the output graph of SYSU-MM01 and added the output graph of the model on the RegDB dataset. Additionally, we have provided appropriate analyses and explanations for all output graphs in the "4.5.2 Output of Visible-Infrared Person Re-ID Results" section.

4 We have added the missing formulas for the Hc-Tri Loss and Enhanced WRT Loss, and provided detailed descriptions for each formula.

5 We have added the embedding position of the Non-Local attention module in our model and provided an illustration to explain it. Other details of our proposed method and some model parameter settings are described in detail in the "4.2. Experimental Settings" section.

Round 2

Reviewer 1 Report

Regarding the article sensors-2387180 titled Joint Modal Alignment and Feature Enhancement for Visible-Infrared Person Re-identification and they have clearly demonstrated the advantages of their approach in comparison to others in the literature.

Authors use a readable English, and some parts are easy to understand. In the same way, they state the justification and main problem.

Furthermore, Authors expose the justification, and I could find the explanation of the main problem in a clearly way.

Then, this paper is based on 39 references from 2015 to 2023 (8 years of research) , and the authors  consider recent references. Only 83% are  from the last five years (from 2018 to 2023) In addition, the references are relevant to the topic they are introducing.

Originality Report shows that this article has a similarity index of 13%, which can be considered as original work. This similarity report is attached to this review.

Please consider the following remarks to improve your article (in some cases, P refers to Page or Pages and L is the Line or Lines where you can find these remarks):

·         Section References is complete, since the most of references  (83%) can be considered as of recent works.

·         The similarity Index of this work is 13%, so it can be considered as original work .

·         The problem and justification are well described.

·         The comparison between state-of-the-art algorithm is complete and enough.

·         Well distribution of the elements to be described or analyzed

·         There is  enought experimentation or comparison of the results that demonstrate the novelty of the project.

·         Results properly exposed

·         Authors use  comparative table of the characteristics of the related work, in addition use some plots that help the experimental results.

·         Equations  are well described or defined

·         Now, this version of the article meets all the requirements to be published. Congratulations to the authors.

So, I suggest modifying, if it is the case, for the publication in the journal Sensors (1424-8220), since the paper by itself have a great potential to publish.

Regards.

Authors use a readable English, and some parts are easy to understand. In the same way, they state the justification and main problem.

Furthermore, Authors expose the justification, and I could find the explanation of the main problem in a clearly way.